# *Troglonectes canlinensis* sp. nov. (Teleostei: Nemacheilidae), a New Troglomorphic Loach from Guangxi, China

**DOI:** 10.3390/ani13101712

**Published:** 2023-05-22

**Authors:** Shu-Jing Li, Jia-Kai Ge, Chun-Yan Bao, Li-Na Du, Fu-Guang Luo, Tong-Xiang Zou

**Affiliations:** 1Key Laboratory of Ecology of Rare and Endangered Species and Environmental Protection, Ministry of Education, Guangxi Normal University, Guilin 541006, Chinazoutongx@126.com (T.-X.Z.); 2Guangxi Key Laboratory of Rare and Endangered Animal Ecology, College of Life Science, Guangxi Normal University, Guilin 541006, China; 3Liuzhou Aquaculture Technology Extending Station, Liuzhou 545006, China; luofuguang@163.com

**Keywords:** taxonomy, complete mitochondrial gene, cave loach, Hongshuihe river

## Abstract

**Simple Summary:**

*Troglonectes* is a small-body loach endemic to the Guangxi and Guizhou provinces of China, showing a particular affinity for cave areas. Twenty species were recorded in this genus, including one new species. The new species, *Tr. canlinensis*, can be distinguished from other congenetic species by their morphological characteristics and molecular evidence. In the genus of *Troglonectes*, the eye, lateral line and scale present or absent, the number of branched pectoral fin rays, caudal fin rays and anal fin rays, and the depth of the upper adipose keel on the caudal peduncle are important identifying characteristics.

**Abstract:**

A new species of the genus *Troglonectes* is described based on specimens from a karst cave in Andong Town, Xincheng County, Liuzhou City, Guangxi, China. *Troglonectes canlinensis* sp. nov. can be distinguished from its congener species by the following combination of characteristics: eye degenerated into a black spot; whole body covered by scales, except for the head, throat, and abdomen; incomplete lateral line; forked caudal fin; 8–10 gill rakers on the first gill arch; 13–14 branched caudal fin rays; 8–9 branched dorsal fin rays; 5–6 anal fin rays; 9–10 pectoral fin rays; upper adipose keel depth mostly 1/2 of the caudal peduncle depth; and caudal fin forked.

## 1. Introduction

Cave loaches of the genus *Troglonectes* Zhang, Zhao, and Yang, 2016 (abbreviation is *Tr*. in this study in order to differ from the abbreviation of *Triplophysa*) are small-bodied fish that mainly occur in the Guangxi and Guizhou provinces of China, showing a particular affinity for cave areas. *Troglonectes* was separated from the genus *Oreonectes* Günther, 1838, that Du et al. [1] divided *Oreonectes* into the *platycephalus* group, i.e., caudal fin rounded or truncated, and the *furcocaudalis* group, i.e., caudal fin forked. Subsequently, Zhang et al. [2] proposed the genus *Troglonectes* and assigned seven nominal species to *Troglonectes*, i.e., *Tr. acridorsalis* (Lan, 2013), *Tr. barbatus* (Gan, 2013), *Tr. elongatus* (Tang, Zhao, and Zhang, 2013), *Tr. macrolepis* (Huang, Du, Chen, and Yang, 2009), *Tr. microphthalmus* (Du, Chen, and Yang, 2008), and *Tr. translucens* (Zhang, Zhao, and Zhang, 2006), in addition to the type species *Tr. furcocaudalis* (Zhu and Cao, 1987). *Troglonectes* can be distinguished from other genera in the Nemacheilidae by possessing narrowly separated nostrils, tube-shaped anterior nostril, tip of the anterior nostril extending into the barbel, dorsal fin origin anterior to the pelvic fin origin, and caudal fin forked or truncated [2]. Except for the species of *Oreonectes* placed in *Troglonectes*, some species of *Paracobitis* and *Triplophysa* were also placed in *Troglonectes* based on their morphology and molecular evidence. Chen et al. [3] described *P. longibarbatus* Chen, Yang, Sket, and Aljancic, 1998 from Libo County, Guizhou, but Du et al. [1] placed it in the *Triplophysa*, based on the morphological characteristics, elongated barbel-like anterior nostril, and sexual dimorphism present in males. Li et al. [4] and Lin et al. [5] described *P. maolanensis* Li, Ran, and Chen, 2006 and *T. jiarongensis* Lin, Li, and Song, 2012 from Guizhou Province, respectively. However, Huang et al. [6] and Luo et al. [7] placed *P. longibarbatus*, *P. maolanensis*, and *T. jiarongensis* in the *Troglonectes*. Subsequently, Luo et al. [7] treated *T. jiarongensis* as a synonym of *Tr. elongatus*. Additionally, Huang et al. [6] mentioned that *O. daqikongensis* Deng, Wen, Xiao, and Zhou, 2016 and *O. shuilongensis* Deng, Wen, Xiao, and Zhou, 2016 also belong to the genus *Troglonectes* due to the forked caudal fin, dorsal fin originating anterior to the pelvic fin origin, and presence of caudal crests. Luo et al. [7] placed *T. huanjiangensis* Yang, Wu, and Lan, 2011, *T. lihuensis* Wu, Yang, and Lan, 2012, *T. lingyunensis* (Liao, Wang, and Luo, 1997), and *O. retrodorsalis* Lan, Yang, and Chen, 1995 in the *Troglonectes* based on molecular analysis. Zhao et al. [8] described one new species, *T. hechiensis* Zhao, Liu, Du, and Luo, 2021, and stated that 17 species were contained within the *Troglonectes*. Luo et al. [7] established one new genus, named *Karstsinnectes* Zhou, Luo, Wang, Zhou, and Xiao, 2023 (type species *Oreonectes anophthalmus* Zheng, 1981), and placed *O. acridorsalis* and *Heminoemacheilus parvus* Zhu and Zhu, 2014 in this genus. In conclusion, 19 species of *Troglonectes* have been recorded in China, including *Tr. barbatus*, *Tr. daqikongensis*, *Tr. donglanensis*, *Tr. dongganensis*, *Tr. duanensis*, *Tr. elongatus*, *Tr. furcocaudalis*, *Tr. hechiensis*, *Tr. huanjiangensis*, *Tr. jiarongensis*, *Tr. lihuensis*, *Tr. lingyunensis*, *Tr. longibarbatus*, *Tr. macrolepis*, *Tr. maolanensis*, *Tr. microphthalmus*, *Tr. retrodorsalis*, *Tr. shuilongensis*, and *Tr. translucens*.

In July 2022, 10 specimens of *Troglonectes* were collected from a cave in Andong Town, Xincheng County, Liuzhou City, Guangxi Zhuang Autonomous Region (hereinafter referred to as Guangxi), China. Morphological and molecular evidence supported these loach specimens representing a new species of *Troglonectes*. Hence, the new species is described herein.

## 2. Materials and Methods

All care and use of experimental animals complied with the relevant laws of the Chinese Laboratory of Animal Welfare and Ethics (GB/T 35892-2018). Specimens of *Troglonectes canlinensis* sp. nov. were collected by F.G. Luo and euthanized rapidly by an overdose of clove oil anesthetic. The right-side pectoral fin and pelvic fin of one specimen were removed and preserved in 99% ethanol. The specimens for the morphological study were stored in 10% formalin, then transferred to 75% alcohol for long-term preservation in the Kunming Natural History Museum of Zoology, Kunming Institute of Zoology (KIZ), Chinese Academy of Science (CAS), China.

Counts and measurements followed Du et al. [1,9], Tang et al. [10], and Lan et al. [11]. Complete mitochondrial genes were sequenced by the Science Corporation of Gene (China) following standard Illumina protocols. Genome sequencing data were submitted to GenBank under Accession No. OQ129618. We retrieved twenty-one complete mitochondrial genomes and five cyt*b* reference sequences of twenty-four Nemacheilidae and two Botiidae species from the NCBI GenBank database for phylogenetic tree reconstruction. *Parabotia fasciata* Dabry de Thiersant, 1872 and *Leptobotia elongata* (Bleeker, 1870), two species of Botiidae, were used as outgroups. To test the phylogenetic position of *Troglonectes canlinensis* sp. nov., Bayesian inference (BI) analysis was performed using MrBayes on XSEDE (v3.2.7a) and the CIPRES Science Gateway [12]. Two runs were performed simultaneously with four Markov chains starting from a random tree. The chains were run for five million generations and sampled every 100 generations. The first 25% of the sampled trees were discarded as burn-in, and the remaining trees were used to create a consensus tree and estimate the Bayesian posterior probabilities (BPPs).

## 3. Results

*Troglonectes canlinensis* sp. nov. (Table 1, Figure 1, Figure 2 and Figure 3)

Holotype. Kunming Natural History Museum of Zoology, KIZ-GXNU202210, 36.0 mm standard length (SL), Andong Town, Xincheng County, Guangxi Zhuang Autonomous Region, China; 24°18.57′ N, 108°59.61′ E, 179 m a.s.l.; collected by F.G. Luo, 20 July 2022.

Paratypes. KIZ-GXNU202207–09, 9 ex., 29.9–54.3 mm SL, collected with holotype.

Diagnosis. *Troglonectes canlinensis* sp. nov., *T. duanensis*, *T. lingyunensis*, *T. macrolepis*, *T. hechiensis*, and *T. retrodorsalis* share their whole trunk being scaled, except for the head and area between the pectoral fins and pelvic fins; other species of *Troglonectes* have scaleless bodies or bodies scaled after the dorsal fin origin in *Tr. furcocaudalis*. However, the new species can be distinguished from *T. duanensis* by the incomplete lateral line (vs. absent), from *T. lingyunensis* and *T. macrolepis* by the eye being present (vs. eye reduced to black pigment), from *T. hechiensis* by the 8–10 inner-gill rakers on first gill arch (vs. 14), and from *T. retrodorsalis* by the tip of the anterior nostril being elongated to barbel-like and the nostril barbel length being nearly twice the nostril tube length (vs. nostril barbel length being less than 1/2 of the tube length).

Description. The morphometric data of the type specimens of *Troglonectes canlinensis* sp. nov. are given in Table 1. Dorsal fin with 4 unbranched and 8–9 branched rays; anal fin with 3 unbranched and 5–6 branched rays; pectoral fin with 1 unbranched and 9–10 branched rays; pelvic fin with 1 unbranched and 5–6 branched rays, caudal fin with 13–14 branched rays; and 8–10 inner-gill rakers on the first gill arch. Vertebrae 4 + 34 (one specimen)

Body elongated, slightly flattened in front, strongly compressed in back. Dorsal profile convex and ventral profile straight in live specimen, but it inversed in preserved specimens. From snout to dorsal fin origin, the body depth increases to its maximum, maximum body depth of 18.2–21.3% SL. Head slightly depressed and flattened, maximum head width greater than the deepest head depth. Anterior and posterior nostrils adjacent, distance less than the diameter of the posterior nostril. Eyes reduced, eye diameter 7.5–11.6% of the lateral head length. Mouth inferior, snout obtuse, upper and lower lips with small furrows and without papillae, median of the lower lip with a V-shaped notch. Three pairs of barbels, inner, outer, and maxillary barbels, extend vertically to the posterior margin of the anterior nostril, anterior margin of the eye, and preopercle, respectively.

Distal margin of dorsal fin truncates, origin anterior to the pelvic fin origin, predorsal length of 54.2–58.6% SL. Tip of pectoral fin reaching halfway to the pelvic fin origin. Tip of pelvic fin far away from the anus. Anus with close-set anal fin base. Caudal fin forked, upper part slightly longer than the lower part. Upper and lower edges of the caudal peduncle with caudal adipose keels, upper adipose keel height mostly 1/2 of the caudal peduncle depth. Caudal peduncle length 90.2–119.0% of its depth. Body trunk covered by tiny scales, except for the ventral surface before the pelvic fin origin. Lateral line incomplete. Cephalic lateral line system with 3 + 3 supratemporal, 6 supraorbital, 3 + 8 infraorbital, and 7–11 preoperculo-mandibular pores.

Stomach U-shaped, intestine long, after stomach, with a bend. Swim bladder divided into two chambers. Anterior chamber covered by dumbbell-shaped bony capsule, and posterior chamber developed.

Coloration. Dorsal surface and trunk of body yellowish brown, abdomen gray and translucent, stomach and intestine visible from outside. Fin membrane hyaline.

Distribution and habitat. The new species was collected from Andong Township, Xincheng County, Laibin City, Guangxi Zhuang Autonomous Region, China (24°18.57′ N, 108°59.61′ E). *Troglonectes canlinensis* sp. nov. lives in a karst cave, where water accumulates to form a pool. Most specimens were collected in the rainy season. During the winter, the pool dries up and the cave opening is too narrow for human access. The water temperature was 20 °C during the survey period in July 2022.

Etymology. The specific name “*canlinensis*” is derived from the pinyin of “can” and “lin”, which refer to resplendence and forest, respectively, with “canlin” symbolizing health and tenacious vitality. *Troglonectes canlinensis* sp. nov. is valuable and rare and requires strong vitality to maintain a viable population. We suggest the common Chinese name “灿 (càn) 林 (lín) 洞 (dòng) 鳅 (qīu)”.

Genetic comparisons. The molecular phylogenies based on BI analysis showed that *Troglonectes* species formed a monophyletic group, sister to the genus *Paranemachilus*. *Troglonectes canlinensis* sp. nov. was sister to the clade including *T. dongganensis*, *T. duanensis*, *T. macrolepis*, *T. microphthalmus*, and *T. translucens*, with bootstrap values of 100. Additionally, the species of *Troglonectes* were divided into two sub-clades: sub-clade 1 contained species with truncated caudal fins, i.e., *Tr. shuilongensis*, *Tr. retrodorsalis*, and *Tr. hechiensis*; sub-clade 2 contains species with forked or emarginated caudal fins, i.e., *Tr. elongatus*, *Tr. jiarongensis*, *Tr. dongganensis*, *Tr. longibarbatus*, *Tr. daqikongensis*, *Tr. barbatus*, *Tr. furcocaudalis*, *Tr. duanensis*, *Tr. donglanensis*, *Tr. microphthalmus*, *Tr. macrolepis*, and *Tr. canlinensis* sp. nov.

Mitochondrial differentiation. The pairwise comparisons of cty*b* revealed that the average uncorrected *p*-distances interspecies of *Troglonectes* ranged from 0.2% to 12.2% (average 7.7%, Table 2). The maximum uncorrected *p*-distance was between *Tr. jiarongensis* and *Tr. barbatus*, and the minimum *p*-distance was between *Tr. translucens* and *Tr. donglanensis*. The average uncorrected *p*-distance between *Tr. canlinensis* sp. nov. and other congeneric species ranged from 3.0% to 9.0% (average 6.8%).
**Identification Key to Species of *Troglonectes***1. Eye present··················································································································································································································2–. Eye degenerated or absent·····················································································································································································52. Body scaled after dorsal fin origin·········································································································································*Tr. furcocaudalis*–. Whole body scaled except for head and thorax··············································································································································33. Caudal fin forked·············································································································································································*Tr. duanensis*–. Caudal fin truncated································································································································································································44. Caudal peduncle length 12.0–13.6% SL··································································································································*Tr. hechiensis*–. Caudal peduncle length 10.8–12.0% SL·····························································································································*Tr. retrodorsalis*5. Eye degenerated with black pigment····························································································································································6–. Eye absent··················································································································································································································106. Body scaleless·············································································································································································································7–. Whole body scaled except for head and thorax··············································································································································87. Upper adipose keel height larger than caudal peduncle depth·············································································*Tr. microphthalmus*–. Upper adipose keel height mostly 1/2 the caudal peduncle depth···········································································*Tr. donglanensis*8. Posterior chamber of swim bladder degenerated············································································································*Tr. lingyunensis*–. Posterior chamber of swim bladder developed·············································································································································99. Total of 12–13 inner gill rakers on first gill arch·····················································································································*Tr. macrolepis*–. Total of 8–10 inner gill rakers on first gill arch······································································································*Tr. canlinensis* sp. nov.10. Caudal fin truncated································································································································································*Tr. shuilongensis*–. Caudal fin emarginated or forked····································································································································································1111. Caudal fin emarginated······················································································································································································12–. Caudal fin forked····································································································································································································1412. Lateral line complete···································································································································································*Tr. jiarongensis*–. Lateral line incomplete or absent······································································································································································1313. Lateral line incomplete································································································································································*Tr. translucens*–. Lateral line absent················································································································································································*Tr. lihuensis*14. Lateral line absent································································································································································································15–. Lateral line complete or incomplete································································································································································1615. Standard length 2.6–3.5 times the lateral head length·······································································································*Tr. barbatus*–. Standard length 4.3–4.9 times the lateral head length································································································*Tr. huanjiangensis*16. Lateral line complete···························································································································································································17–. Lateral line incomplete·························································································································································································1917. Dorsal fin with six branched rays, anal fin with four branched rays······································································*Tr. maolanensis*–. Dorsal fin with eight or nine branched rays, anal fin with six branched rays···················································································1818. Standard length 10.1–14.0 times the caudal peduncle depth··················································································*Tr. daqikongensis*–. Standard length 14.5–18.1 times the caudal peduncle depth····················································································*Tr. longibarbatus*19. Pelvic fin origin opposite the dorsal fin origin···············································································································*Tr. dongganensis*–. Pelvic fin origin anterior to the dorsal fin origin······················································································································*Tr. elongatus*

## 4. Discussion

The genus *Troglonectes* is currently distributed in the Pearl River system in Guangxi and Guizhou Provinces, and is endemic to China. Although Zhang et al. [2] mentioned that one of the identifying features of the genus is a forked caudal fin, there are truncated, emarginated, and forked caudal fins, three types of caudal fin. The phylogenetic tree indicates that the species of *Troglonectes* divided into sub-clade 1 contains species with truncated caudal fins and sub-clade 2 contains species with emarginated or forked caudal fins. Hence, the caudal fin shape and phylogenetic tree support that *Troglonectes* could be divided into two groups; the truncated caudal fin group contains *Tr. hechiensis*, *Tr. retrodorsalis*, and *Tr. shuilongensis*, and the emarginated or forked caudal fin group contains *Tr. donglanensis*, *Tr. microphthalmus*, *Tr. macrolepis*, *Tr. canlinensis* sp. nov., *Tr. lingyunensis*, *Tr. barbatus*, *Tr. huanjiangensis*, *Tr. longibarbatus*, *Tr. maolanensis*, *Tr. daqikongensis*, *Tr. dongganensis*, *Tr. elongatus*, *Tr. translucens*, *Tr. jiarongensis*, *Tr. lihuensis*, *Tr. furcocaudalis*, and *Tr. duanensis*. Thus, based on our BI analysis and external characteristics, the genus description for *Troglonectes* includes the following characteristics: anterior and posterior nostrils separated by a short distance shorter than the diameter of the posterior nostril, tip of anterior nostril elongated to barbel-like, and adipose keels on the upper and lower edges of the caudal peduncle present.

Luo et al. [7] treated *Tr. donglanensis* and *Tr. duanensis* as synonyms of *Tr. translucens*, and *Tr. jiarongensis* and *Tr. dongganensis* as synonyms of *Tr. elongatus* based on morphological characteristics and a lack of genetic differences, respectively. *Troglonectes dongganensis*, *Tr. elongatus*, *Tr. jiarongensis*, and *Tr. longibarbatus* formed a monophyletic group in the phylogenetic tree, and the genetic distance was 0.4–1.0% (average 0.7%). However, they can be morphologically distinguished from each other by the lateral line (complete in *Tr. jiarongensis* and *Tr. longibarbatus*, incomplete in *Tr. elongatus* and *Tr. dongganensis*, and absent in *Tr. huanjiangensis*), branched caudal fins (16 in *Tr. jiarongensis* and 13–14 in other species), and body depth (8.6–10.7% SL in *Tr. elongatus*, and more than 13% in *Tr. dongganensis*, *Tr. huanjiangensis*, *Tr. jiarongensis*, and *Tr. longibarbatus*). Hence, we treated *T. dongganensis*, *T. elongatus*, *T. huanjiangensis*, *T. jiarongensis*, and *T. longibarbatus* as valid species in this study. Additionally, *Tr. donglanensis*, *Tr. duanensis*, and *Tr. translucens* can be distinguished from each other by the 16 branched caudal fins in *Tr. translucens* (vs. 13–14 in *Tr. donglanensis* and *Tr. duanensis*) and the body being covered by scales and the lateral line being absent in *Tr. duanensis* (vs. scaleless and incomplete lateral line in *Tr. donglanensis* and *Tr. translucens*). Thus, we propose *Tr. donglanensis*, *Tr. duanensis*, and *Tr. translucens* as valid species.

Within the genus *Troglonectes*, 20 valid species were recorded, including the new species. *Troglonectes canlinensis* sp. nov. can be distinguished from *Tr. hechiensis*, *Tr. retrodorsalis*, and *Tr. shuilongensis* by its forked caudal fin (vs. truncated) and upper adipose keel height being mostly 1/2 of the caudal peduncle depth (vs. 1/4), and it can be further distinguished from *Tr. shuilongensis* by its degenerated eye with a black pigment (vs. absent), scaled body (vs. scaleless), incomplete lateral line (vs. complete), and 8−10 inner-gill rakers on the first gill arch (vs. 10−12); from *Tr. hechiensis* by the 8−10 inner-gill rakers on the first gill arch (vs. 14) and 9−10 branched pectoral fin rays (vs. 11); and from *Tr. retrodorsalis* by the 8−10 inner-gill rakers on the first gill arch (vs. 13−14) and 9−10 branched pectoral fin rays (vs. 11−12). *Troglonectes canlinensis* sp. nov. is different from *Tr. translucens*, *Tr. jiarongensis*, and *Tr. lihuensis* owing to its forked caudal fin (vs. emarginated) and scaled body (vs. scaleless); it can be further differentiated from *Tr. jiarongensis* and *Tr. lihuensis* by its incomplete lateral line (vs. absent in *Tr. lihuensis* and complete in *Tr. jiarongensis*) and upper adipose keel height being mostly 1/2 of the caudal peduncle depth (vs. equal to the caudal peduncle depth); and from *Tr. jiarongensis* by the 13−14 branched caudal fin rays (vs. 16). The new species is different from *Tr. furcocaudalis* owing to its whole body being scaled, except for the head and thorax (vs. scaled after the dorsal fin origin), 8−10 inner-gill rakers on the first gill arch (vs. 12−13), and 5−6 branched pelvic fin rays (vs. 7); from *Tr. duanensis* owing to the incomplete lateral line (vs. absent), 8−10 inner-gill rakers on the first gill arch (vs. 13), anal fin with 5−6 branched rays (vs. 6−7), and eye degenerated with black pigment (vs. present); from *Tr. lingyunensis* by the developed posterior chamber of the swim bladder (vs. degenerated), caudal fin with 13−14 branched rays (vs. 16), dorsal fin with 8−9 branched rays (vs. 6−7), and upper adipose keel height being mostly 1/2 half of the caudal peduncle depth (vs. 1/4); and from *Tr. macrolepis* by the 8−10 inner-gill rakers on the first gill arch (vs. 12−13), dorsal fin with 8−9 branched rays (vs. 9−11), pectoral fin with 9−10 branched rays (vs. 10−12), and upper adipose keel height being mostly 1/2 of the caudal peduncle depth (vs. equal with caudal peduncle depth). *Troglonectes canlinensis* sp. nov. is different from *Tr. barbatus*, *Tr. huanjiangensis*, *Tr. longibarbatus*, *Tr. maolanensis*, *Tr. daqikongensis*, *Tr. dongganensis*, *Tr. elongatus*, *Tr. donglanensis*, and *Tr. microphthalmus* owing to its scaled body (vs. scaleless); it can be further distinguished from these species by the eye being degenerated with black pigment (vs. absent in *Tr. barbatus*, *Tr. huanjiangensis*, *Tr. longibarbatus*, *Tr. maolanensis*, *Tr. daqikongensis*, *Tr. dongganensis*, and *Tr. elongatus*), lateral line being incomplete (vs. complete in *Tr. longibarbatus*, *Tr. maolanensis*, and *Tr. daqikongensis*, or absent in *Tr. barbatus* and *Tr. huanjiangensis*), dorsal fin having 8−9 branched rays (vs. 10−11 in *Tr. microphthalmus* and 6 in *Tr. maolanensis*), anal fin having 5−6 branched rays (vs. 4 in *Tr. maolanensis* or 6−7 in *Tr. huanjiangensis*, *Tr. longibarbatus*, *Tr. daqikongensis*, *Tr. dongganensis*, *Tr. elongatus*, *Tr. donglanensis*, and *Tr. microphthalmus*), and upper adipose keel height being mostly 1/2 of the caudal peduncle depth (vs. equal to the caudal peduncle depth in *Tr. barbatus*, *Tr. huanjiangensis*, *Tr. longibarbatus*, *Tr. maolanensis*, *Tr. daqikongensis*, *Tr. dongganensis*, *Tr. elongatus*, and *Tr. microphthalmus*).

Species of *Troglonectes* are highly adapted to survive in cave habitats and are found only in limited regions with relatively small populations. Ma et al. [13] mentioned that cave fish have morphological adaptations to extreme cave environments, including the degeneration or disappearance of the eyes, reduced pigment, and scales. Additionally, cave fish have specialized features including well-developed tentacles and prolonged pectoral fins. Species of *Troglonectes* have developed barbels, well-developed adipose keel on the upper and lower caudal peduncles, reduced or no eyes, lateral line, scales, and pigment; these characteristics are adaptions to cave environments. As their life histories are limited to caves, these fish are vulnerable to various threats, such as habitat degradation, hydrological alterations, environmental pollution, resource overexploitation, and non-native species introduction [13]. Karst caves and subterranean streams are common geological features in Guangxi. More than 300 freshwater fish species have been recorded in Guangxi, including 61 cavefish [11]. On 16 September 2022, the Department of Forestry of the Guangxi Zhuang Autonomous Region published a list of wildlife under key protection in Guangxi, which included all cavefish species. The new species is currently only known from the type locality, where a few specimens were collected when the water rose from the cave during the rainy season. The discovery of this previously unknown species can hopefully lead to conservation measures to protect this area.

## 5. Conclusions

One new species of *Troglonectes* is described herein based on its morphological characteristics and molecular analysis. Additionally, the phylogenetic tree indicates species of *Troglonectes* divided into two sub-clades, viz. the truncated caudal fin sub-clade and emarginated or forked caudal fin sub-clade.

## 6. Nomenclatural Acts Registration

This published work and the nomenclatural acts it contains have been registered in ZooBank LSIDs (Life Science Identifiers) and can be resolved, and the associated information can be viewed through any standard web browser by appending the LSID to the prefix http://zoobank.org/ (accessed on 10 February 2023).

Publication LSID:

urn:lsid:zoobank.org:pub:5DB60B6B-94EC-4E18-B734-9258F9E31D2A.

*Troglonectes canlinensis* LSID:

urn:lsid:zoobank.org:act:1FEADE1C-2EBF-4956-A052-F6926AB9124C.

## Figures and Tables

**Figure 1 animals-13-01712-f001:**
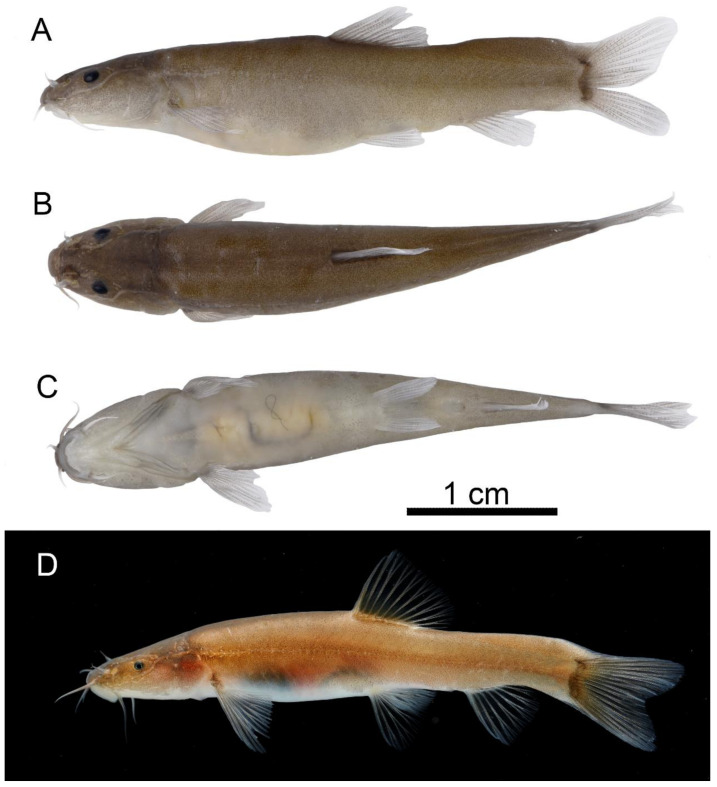
Preserved and living photos of *Troglonectes canlinensis* sp. nov. Holotype KIZ-GXNU202210, (**A**) lateral view; (**B**) dorsal view; (**C**) ventral view; (**D**) living photo. Scale = 1 cm.

**Figure 2 animals-13-01712-f002:**
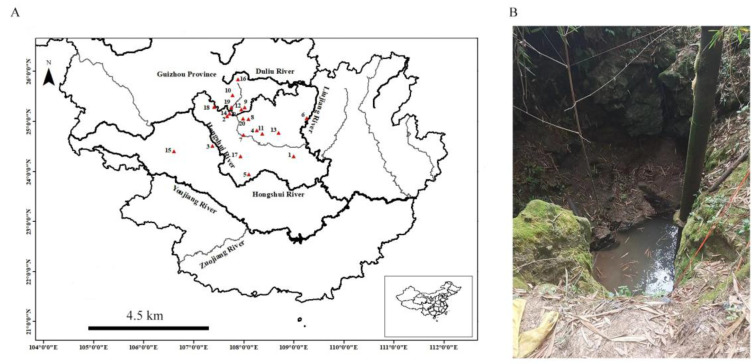
(**A**) Collection site of species of *Troglonectes* (red triangle). 1. *Tr. canlinensis*; 2. *Tr. barbatus*; 3. *Tr. donglanensis*; 4. *Tr. dongganensis*; 5. *Tr. duanensis*; 6. *Tr. furcocaudalis*; 7. *Tr. hechiensis*; 8. *Tr. huanjiangensis*; 9. *Tr. jiarongensis*; 10. *Tr. longibarbatus*; 11. *Tr. macrolepis*; 12. *Tr. maolanensis*; 13. *Tr. microphthalmus*; 14. *Tr. lihuensis*; 15. *Tr. lingyunensis*; 16. *Tr. shuilongensis*; 17. *Tr. translucens*; 18. *Tr. retrodorsalis*; 19. *Tr. daqikongensis*; 20. *Tr. elongatus*. (**B**) habitat of *Troglonectes canlinensis* sp. nov. in Guangxi Zhuang Autonomous Region, China.

**Figure 3 animals-13-01712-f003:**
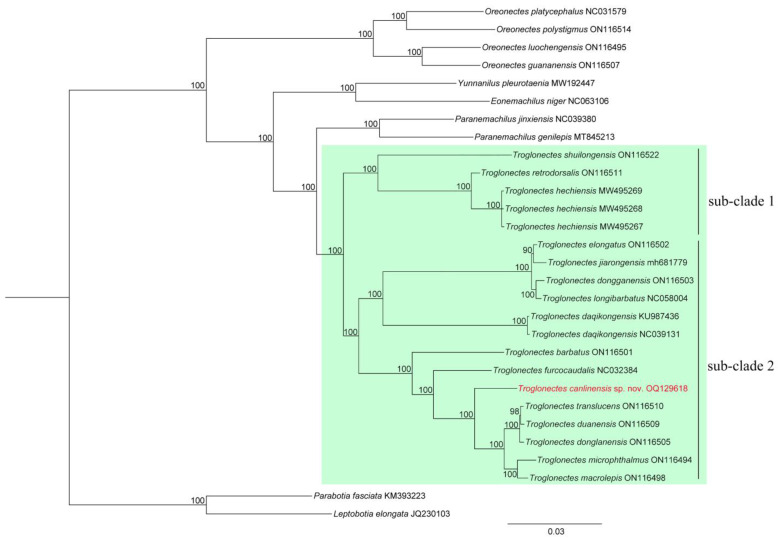
Bayesian phylogram of *Troglonectes* based on the mitochondrial genomes of 24 nemacheilid species and 2 botiid species (outgroups). Numbers above branches are BPPs.

**Table 1 animals-13-01712-t001:** Morphometric and meristic data of *Troglonectes canlinensis* sp. nov. The range, mean, and standard deviation (mean ± SD) include holotype values.

	Holotype	Range (*n* = 10)	Mean ± SD
Total length (mm)	43.0	35.6–65.3	46.1 ± 8.6
Standard length (mm)	36.0	29.9–54.3	38.7 ± 7.3
Percentage of standard length (%)			
Body depth	22.0	18.2–22.0	19.8 ± 1.1
Lateral head length	28.0	25.8–29.6	27.7 ± 1.1
Predorsal length	57.8	54.2–58.7	56.9 ± 1.2
Prepelvic length	61.0	57.8–61.2	59.8 ± 1.1
Preanal length	80.0	75.6–81.3	79.3 ± 1.7
Preanus length	75.7	74.1–77.4	75.9 ± 1.2
Caudal peduncle length	11.5	10.9–14.6	12.8 ± 1.3
Caudal peduncle depth	13.9	11.3–13.9	12.2 ± 0.7
Head width	18.1	16.0–18.1	16.9 ± 0.8
Pectoral fin length	16.1	15.3–16.7	16.1 ± 0.4
Pelvic fin length	11.4	11.4–14.3	12.9 ± 0.8
Percentage of lateral head length (%)			
Eye diameter	10.3	7.5–11.6	10.1 ± 1.1
Interorbital width	35.81	26.5–39.4	32.4 ± 4.4
Snout length	35.5	34.6–46.6	40.5 ± 3.4
Head width	64.5	54.2–67.6	61.3 ± 4.0
Head depth	51.6	46.9–55.9	51.3 ± 2.6
Maxillary barbel length	28.6	19.3–37.1	29.1 ± 5.4
Outer barbel length	26.7	21.9–36.3	29.7 ± 3.9
Inner barbel length	15.8	12.0–19.5	17.4 ± 2.2
Percentage of caudal peduncle length (%)			
Caudal peduncle depth	120.5	84.1–120.5	96.5 ± 12.5
Dorsal fin rays	4, 9	4, 8–9	
Pectoral fin rays	1, 9	1, 9–10	
Pelvic fin rays	1, 6	1, 5–6	
Anal fin rays	3, 6	3, 5–6	
Caudal fin rays	14	13–14	

**Table 2 animals-13-01712-t002:** Interspecific genetic distances (uncorrected p-distance) between pairs of *Troglonectes* species based on cytochrome *b* mtDNA sequences.

		1	2	3	4	5	6	7	8	9	10	11	12	13	14	15
1	*Tr. barbatus*	-														
2	*Tr. canlinensis*	0.075														
3	*Tr. daqikongensis*	0.105	0.090													
4	*Tr. dongganensis*	0.119	0.099	0.107												
5	*Tr. donglanensis*	0.072	0.033	0.101	0.096											
6	*Tr. duanensis*	0.071	0.033	0.102	0.095	0.003										
7	*Tr. elongatus*	0.118	0.098	0.108	0.010	0.093	0.092									
8	*Tr. furcocaudalis*	0.066	0.048	0.096	0.098	0.046	0.043	0.097								
9	*Tr. hechiensis*	0.099	0.098	0.102	0.098	0.085	0.084	0.097	0.084							
10	*Tr. jiarongensis*	0.122	0.098	0.107	0.010	0.098	0.097	0.007	0.101	0.096						
11	*Tr. longibarbatus*	0.117	0.095	0.103	0.004	0.092	0.091	0.006	0.096	0.094	0.006					
12	*Tr. macrolepis*	0.068	0.030	0.099	0.097	0.016	0.017	0.096	0.047	0.084	0.098	0.093				
13	*Tr. microphthalmus*	0.069	0.031	0.098	0.098	0.014	0.014	0.097	0.046	0.086	0.099	0.094	0.008			
14	*Tr. retrodorsalis*	0.097	0.079	0.099	0.098	0.085	0.084	0.097	0.082	0.012	0.095	0.094	0.081	0.082		
15	*Tr. shuilongensis*	0.115	0.099	0.109	0.100	0.101	0.100	0.097	0.101	0.089	0.101	0.096	0.102	0.098	0.085	
16	*Tr. translucens*	0.072	0.031	0.100	0.094	0.002	0.003	0.091	0.046	0.083	0.096	0.090	0.016	0.014	0.083	0.100

## Data Availability

The morphometric and meristic data of *Troglonectes canlinensis* sp. nov. in this study have been deposited in the (ScienceDB) repository (https://www.scidb.cn/s/uaI3yu (accessed on 20 March 2023). The photo and phylogenetic tree data used to support the findings of this study are included within the article.

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
