# Peer review of "Troglonectes canlinensis* sp. nov. (Teleostei: Nemacheilidae), a New Troglomorphic Loach from Guangxi, China"

_animals, 2023, doi:10.3390/ani13101712_

Round 1
Reviewer 1 Report
This is a very well-written and well-presented manuscript.
The main question addressed by the research is Definition and description of a new species of Troglonectyes, a cavefish, from southern China. The meaning is clear throughout and I have no suggestions for improvement apart from one minor grammar issue: line41-42, 'T. agridorsalis is not a species of Troglonectes.....'
New species are always worthy of publication, and this one has specific adaptations to an unusual habitat, so I believe the information is worthy of publication.
The methodology with both morphological and genetic data is of high standard. Adequate controls (other species for comparison) are already present in the analysis. A wider discussion of adaptations of cavefish and their unusual morphologies would be possible, and interesting, but this would be best as a separate paper. I can see no areas needing improvement.
Figure 2 might need some modification to make it better suited to production : if the coloured maps, upper part of figure, were made larger, and the black and white inset, below, made smaller, the very small fonts in the coloured images may be better legible.
Author Response
Response to the Reviewers
We are grateful for the excellent queries and suggestions provided by the reviewers. We think these comments largely improve the quality of our manuscript. All comments about languages are either correct suggested or reconstructed since many relevant section have been rewritten. The language is polished throughout the manuscript and will be refined after the main content being accepted. Here, we provide our point-to-point response to comments in below:
- The main question addressed by the research is Definition and description of a new species of Troglonectyes, a cavefish, from southern China. The meaning is clear throughout and I have no suggestions for improvement apart from one minor grammar issue: line41-42, 'T. agridorsalis is not aspecies of Troglonectes.....'
Response: accepted, “a” was added in front of species.
- Figure 2 might need some modification to make it better suited to production: if the coloured maps, upper part of figure, were made larger, and the black and white inset, below, made smaller, the very small fonts in the coloured images may be better legible.
Response: Figure 2 has been changed.
Reviewer 2 Report
Dear authors and editor,
The manuscript is an important contribution to the knowledge of the genus Troglonectes. However, I believe that the authors have very relevant and little explored results at hand (e.g. phylogeny results). The authors cite two works that are unpublished Master Thesis and therefore are not considered publications I strongly recommend that they are not used in the text.
In general, these are my suggestions (I left everything marked throughout the PDF):
- Modify the title, as the name of the genre is repeated.
"Troglonectes canlinensis sp. nov. (Teleostei: Nemacheilidae), a new troglomorphic loach from Guangxi, China"
- Leave the keywords in alphabetical order and change the Guangxi that is already in the title.
- Authors can improve the introduction that has disconnected sentences, I recommend trying to leave it in the following order:
Who is the genus Troglonectes, how did this genus arise, what are its characteristics, what are the groups of species. (In that order, the beginning of the introduction would be better).
- I recommend removing the part of the text that is starting on line 39 about Wang's Thesis.
- Line 87 the diagnosis is incomplete, where are the 17 species being compared? Why are some in remarks and not here?
- Table: a subtitle must be inserted or a new table must be created, describing what each letter of the meristic counts means (V is wrong, I recommend changing it to Pv or Plv.)
Figure 2 : The map is not being explored well. I would like to see where the other species are distributed in that area as well, it would be much more informative.
Genetic comparisons.
The result here is very interesting and little explored, it could be indicated, for example, the supports for the new species.
Discussion:
Here I believe is where there would be more modifications to be made. The authors have to talk about their results here about new diagnosis, it is being done based on their results, this must be indicated.
I recommend deleting the text part of line 151, due to it being an unpublished master thesis. Looking at the Eschmeyer's Catalog of Fishes website, all species are still considered valid (since only what is duly published should be followed). With this, the authors can and should explore the molecular result obtained by them and discuss based on this result.

Author Response
Response to the Reviewers
We are grateful for the excellent queries and suggestions provided by the reviewers. We think these comments largely improve the quality of our manuscript. All comments about languages are either correct suggested or reconstructed since many relevant section have been rewritten. The language is polished throughout the manuscript and will be refined after the main content being accepted. Here, we provide our point-to-point response to comments in below:
- The manuscript is an important contribution to the knowledge of the genus Troglonectes. However, I believe that the authors have very relevant and little explored results at hand (e.g. phylogeny results). The authors cite two works that are unpublished Master Thesis and therefore are not considered publications I strongly recommend that they are not used in the text.
Response: master thesis was changed to publication, Luo et al. (2023)
- In general, these are my suggestions (I left everything marked throughout the PDF):
Response: these suggestion had been accepted and manuscript changed.
- Modify the title, as the name of the genre is repeated.
"Troglonectes canlinensis sp. nov. (Teleostei: Nemacheilidae), a new troglomorphic loach from Guangxi, China"
Response: accepted, the title was changed
- Leave the keywords in alphabetical order and change the Guangxi that is already in the title.
Response: Guangxi was delete, complete mitochondrial gene was added.
- Authors can improve the introduction that has disconnected sentences, I recommend trying to leave it in the following order:
Who is the genus Troglonectes, how did this genus arise, what are its characteristics, what are the groups of species. (In that order, the beginning of the introduction would be better).
Response: this part was rewritten according to reviewer’s suggestion.
- I recommend removing the part of the text that is starting on line 39 about Wang's Thesis.
Response: delete the master thesis.
-. Line 50 the cave have a name?
Response: the cave without name.
-. Line 80 from all specimens?
Response: No, only one specimen was cut fins.
- Line 87 the diagnosis is incomplete, where are the 17 species being compared? Why are some in remarks and not here?
Response: the diagnosis was changed, it can be distinguished from 19 species.
- Table: a subtitle must be inserted or a new table must be created, describing what each letter of the meristic counts means (V is wrong, I recommend changing it to Pv or Plv.)
Response: D, P, V, C, and A were explained.
Figure 2 : The map is not being explored well. I would like to see where the other species are distributed in that area as well, it would be much more informative.
Response: Figure 2 was changed.
Discussion:
Here I believe is where there would be more modifications to be made. The authors have to talk about their results here about new diagnosis, it is being done based on their results, this must be indicated.
Response: more information were provided in discussion. And a key of Troglonectes were add in result.
I recommend deleting the text part of line 151, due to it being an unpublished master thesis. Looking at the Eschmeyer's Catalog of Fishes website, all species are still considered valid (since only what is duly published should be followed). With this, the authors can and should explore the molecular result obtained by them and discuss based on this result.
Response: unpublished master thesis was delete.